# ODE PARAMETER IDENTIFICATION: AN INTEGRAL MATCHING APPROACH

## ABSTRACT

We present a novel method to identify parameter of nonlinear Ordinary Differential Equations (ODEs) using time series data. Our approach fits parameters by matching a collocation-based estimate of the integral of the learned derivative to an interpolation of the trajectory, thus avoiding the computational cost of ODE solvers in adjoint methods and the sensitivity to noise of derivative estimates in gradient matching methods. By employing batching strategies based on time subintervals and state components, our method achieves linear complexity in relation to system dimensions and dataset sizes. The method is highly parallel enabling fast gradient evaluations and a faster convergence than adjoint methods. For fully observed systems, we demonstrate the method on canonical dynamical systems, where the method achieves speed-ups of three orders of magnitude over adjoint methods and an increased robustness against observational noise. We provide an extension to partially observed systems and demonstrate the method on the Lorenz63 attractor.

## 1 INTRODUCTION

Ordinary Differential Equations (ODEs) are widely used to model dynamical systems in fields such as physics, biology, and engineering, Strogatz (2014); Villaverde and Banga (2014). Estimating unknown parameters of arbitrary nonlinear ODEs -derived from physical laws, or postulated like SIR models in epidemiology- from noisy, partially observed time-series data is important.

### 1.1 PROBLEM STATEMENT

Let $\mathbf{x}(t) \in \mathbb{R}^n$ denote the state of a system of dimension $n$ at time $t$. We are given $M$ noisy, partial observations $\mathcal{O}\mathbf{x}(t_m)$ from a single trajectory, where $t_m \in [0, T]$ for all $m = 1, \ldots, M$, and $\mathcal{O}$ projects the state onto the first $p \leq n$ components. We consider the inverse problem of determining the optimal parameters $\mathbf{\Theta}^*$ and initial condition $\mathbf{X}_0^*$ that minimize the mean squared error between the observations and $\hat{\mathbf{x}}(t)$, the trajectory obtained by integrating the ODEs (1a) defined by a parameterized function $f$ with these parameter from the initial condition. With Newton's notation $\dot{\mathbf{x}} = \frac{d\mathbf{x}}{dt}$:

$$\mathbf{\Theta}^*, \mathbf{X}_0^* = \arg \min_{\mathbf{\Theta}, \mathbf{X}_0} \quad \frac{1}{M} \sum_{m=1}^{M} \|\mathcal{O}\hat{\mathbf{x}}(t_m) - \mathcal{O}\mathbf{x}(t_m)\|^2$$

$$\text{s.t.} \quad \dot{\hat{\mathbf{x}}}(t) = f(\hat{\mathbf{x}}(t), \mathbf{\Theta}), \quad \forall t, \tag{1a}$$

$$\hat{\mathbf{x}}(0) = \mathbf{X}_0. \tag{1b}$$

**Note:** This setting includes dynamics that are polynomial in the state and dynamics with nonlinearities on the unknown parameters, encountered in physics, when estimating the parameters of chemical kinetics, and prior-less settings such as Neural ODEs Chen et al. (2018), where a neural network of weights $\mathbf{\Theta}$ acts as a universal approximator. It handles asynchronous measurements and missing data.

### 1.2 CONTRIBUTIONS

This problem, known as system identification, has received considerable attention in optimal control and scientific machine learning, see Åström and Eykhoff (1971); Söderström and Stoica (1989); Ljung

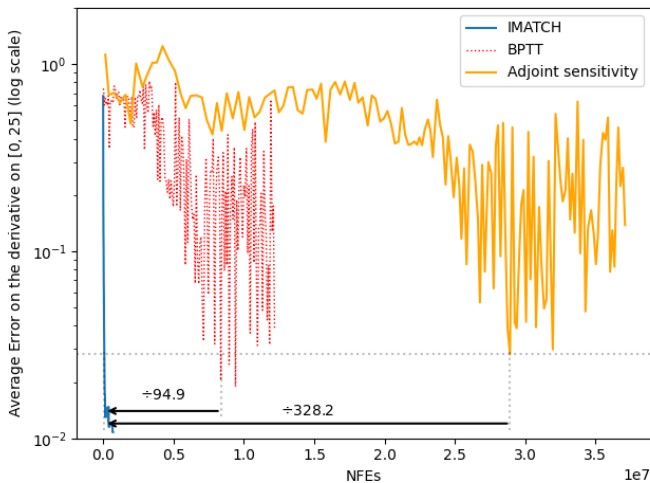

Figure 1: On the damped oscillator from Chen et al. (2018), our algorithm (blue) fits a neural ODE with fewer network evaluations and greater accuracy than Backpropagation through time (BPTT) (red) and Adjoint sensitivity (orange). Our method achieves in 2.5s the best accuracy reached by the adjoint sensitivity within 15 minutes on CPU. The number of function evaluations is divided by respectively 95 and 328 compared to BPTT and Adjoint and computation times by factors 50 to 450.

(1999). Parameter estimation in nonlinear ODEs is challenging due to the complexity of dynamics and the nonconvex optimization landscape of the learning problem, Varah (1982). Data noise, irregular sampling, asynchronous measurements, and unobserved dimensions further complicate the issue. Existing approaches to this problem can be broadly categorized into ODE solver-based methods and surrogate methods. The former integrate ODEs but are computationally expensive, especially for large dimension systems as detailed later while the latter including gradient matching Varah (1982); Ramsay et al. (2007); Poyton et al. (2006); Calderhead et al. (2008); Dondelinger et al. (2013) approximate ODE solutions or their derivatives, trading accuracy for computational efficiency.

In this paper, we introduce and study a novel surrogate approach that matches a numerical integration of learned derivatives to an interpolation of the trajectory for the fully observed case, and its extension to partially observed systems. Our main contributions are:

1. **Speed and Robustness:** On classical benchmarks, for the fully observed case, the proposed method is more computationally efficient than ODE-solver methods, being up to three orders of magnitude faster, and more robust to noise on observations as it avoids the noise-sensitive estimation of temporal derivatives from noisy data. Batching strategies enable parallel processing of different dimensions of the state and time subintervals and the learning of systems of high dimension, as demonstrated in the model Lorenz (1996).

2. **Theoretical guarantees:** For the fully observed case, we show bounds between the optimum of Problem 1 and the loss optimized by the proposed method.

3. **Partially Observed Systems:** We extend our method to handle partially observed systems, demonstrating its ability to estimate initial conditions and ODEs for unobserved dimensions.

**Plan:** In Section 2, we detail related methods and their tradeoffs. We first present the algorithm, its derivation and theoretical result in the case of fully observed systems in Section 3 with numerical experiments in Section 4. We then present an extension to partially observed systems in Section 5.

## 2 BACKGROUND AND RELATED WORK

To better position the method in Section 3, we first detail existing literature that is relevant to our case of continuous time data, focusing on ODE Solvers and surrogate methods.

**Direct approaches** use numerical integration to estimate the gradient of the loss on parameters, with different trade-offs between memory, accuracy, and complexity. Most accurate, the continuous

Table 1: Efficiency comparison: Function Evaluations and memory. Our method is more computationally efficient with a controlled memory overhead. For on a numerical tolerance of $\epsilon$, an explicit Runge-Kutta method of order $K$ uses $N(K,T) = \left(\mathcal{O}(\epsilon^{-1/K}T)\right)$ steps; our method $\tilde{N} = \left(\mathcal{O}(\epsilon^{-1/2K-1}T)\right)$.

| METHOD | ADAPT. | STIFF | #NFE | MEMORY | ACCURACY Fwd/Bwd | REF |
|---|---|---|---|---|---|---|
| ADJOINT-RK | ✓ | | $4KN$ | $O(n)$ | ↑ \| ↓ | CHEN ET AL. (2018) |
| BPTT-RK | ✓ | ✓ | $2KN$ | $\mathcal{O}(nN)$ | ↑ \| ↑ | GRUSLYS ET AL. (2016) |
| ACA-CVODE | ✓ | ✓ | $4KN$ | $\mathcal{O}(nN)$ | ↑ \| ↑ | ZHUANG ET AL. (2020) KIM ET AL. (2021) |
| LTC | | ✓ | $8N(4,T)$ | $\mathcal{O}(nN(4,T))$ | ↑ \| ↑ | HASANI ET AL. (2020) |
| THIS PAPER | | ✓ | $2K\tilde{N}$ | $\mathcal{O}(nK)$ | ↑ \| ↑ | |

version of the Recurrent Neural Networks (RNNs) method -Backpropagation Through Time (BPTT)- is obtained by differentiating through an ODE Solver using frameworks such as Pytorch Paszke et al. (2019) or JAX Bradbury et al. (2018), or through a custom ODE Solver as in Forgione and Piga (2021) and Hasani et al. (2020) for Liquid Time-Constant (LTC) networks. It however requires memory to store each step of the forward integration for the backpropagation. Based on the Pontryagin principle Pontryagin et al. (1962), the *adjoint sensitivity method* solves this memory issue by estimating gradients using a backwards integration, storing only the terminal value of the state, see Chen et al. (2018); Rubanova et al. (2019); Gholami et al. (2019). As discrepancies between the forward and backward integrations reduce the accuracy of gradients for this last method, adaptive checkpointing (ACA) in Zhuang et al. (2020) uses checkpoints to enable a backward shooting method for the adjoint on smaller subtrajectories, and Kim et al. (2021) stores the forward pass while using a backwards integration of the adjoint. Direct methods are computationally expensive, and sequential being autoregressive. They are sensitive to initialization: inaccuracies on the initial condition hamper the accuracy of gradients on parameters and poor parameter initialization or bifurcations can lead to an unpredictable number of adaptive steps used to control numerical error in the ODE Solver Hairer et al. (1993). Table 1 compares the memory and computational complexity of direct methods with the method in this paper, wall-clock times comparisons are included in section 4.

**Surrogate methods:** To avoid numerical integration, gradient matching, introduced in Varah (1982), fits parameters $\Theta$ to match an estimate of derivative obtained by finite differences, see also Ramsay et al. (2007); Tjoa and Biegler (1991); Niu et al. (2016). As estimating derivatives is sensitive to noise on data, Roesch et al. (2021) uses local smoothing techniques to estimate the trajectory and its derivative. The Sparse Identification of Nonlinear Dynamics (SINDy) framework, introduced in Brunton et al. (2016) combines gradient matching with sparse regression when $f$ is a linear combination of nonlinear functions. Weak formulations and integral form using trapezoidal integration for regularly sampled data are presented in Messenger and Bortz (2021); Schaeffer and McCalla (2017). Weak forms are more robust to noise on observations, but are not generally tractable for arbitrary dynamics $f$. Calderhead et al. (2008); Dondelinger et al. (2013) have explored Bayesian approaches to combine gradient matching with sampling strategies and Bayesian updates. We show that using collocation methods that smoothing with particular polynomials leads to guarantees on the numerical integration.

## 2.1 COLLOCATION METHODS

Collocation methods have become increasingly popular to solve optimal control problems; see Betts (2010). These methods are implicit integration methods where the value of state and control (the parameters $\Theta$) at specific discretization nodes are decision variables of a nonlinear program. Although, as noted in Varah (1982), these methods are the backbone of gradient matching, their direct use for system identification is original. We selected the Legendre-Gauss-Radau (LGR) approach for its suitability to initial value problems and its properties: it is A-stable, i.e., with numerical stability guarantees for classes of initial value problems, symplectic, i.e., preserving the Hamiltonian of the system, and has an approximation error is $o(h^{2K-1})$, where $K$ is the degree of the approximating polynomial and $h$ is the size of the time step, see Fahroo and Ross (2008); Garg et al. (2011b) for discussions and proofs. While other collocation methods are compatible with our approach, our choice ensures that the KKT conditions discretize the Pontryagin Principle, which connects our method to the adjoint sensitivity method, see Wei et al. (2016).

## 3 ALGORITHM FOR FULLY OBSERVED SYSTEMS

We present *Integral Matching*, in pseudo-code (Algorithm 1), a method alternating interpolations of the trajectory and gradient descents on $\ell(\boldsymbol{\Theta}, \mathbf{X})$, a collocation-based estimation of:

$$\sum_{i=1}^{N-1} \sum_{j=1}^{K} \| \int_{t_i}^{t_i+h_i\tau_j} \dot{\mathbf{x}}(t) - \mathbf{f}(\mathbf{x}(t), \boldsymbol{\Theta})dt \|^2$$

where $h_i = t_{i+1} - t_i$, $(\tau_j)$ are collocation nodes, $\mathbf{x}(t)$ a polynomial approximation of the state given by $\mathbf{X}$ its values at nodes. Section 3.1 derives the loss $\ell$ and Section 3.3 studies the algorithm.

---

**Algorithm 1** Integral Matching

1: **Input:** data $(t_m, \mathbf{x}(t_m))_{m=1,\ldots,M}$, order $K$, subinterval length $h$, initialization of $\boldsymbol{\Theta}$
2: **Build denoised set** $F=\{(t_f, \mathbf{x}_f(t_f)\}$
3: **repeat**
4:     **generate** a set $S$ of subintervals $[a,a+h]$
5:     **for** $s$ **in** $S$ **do**
6:         **compute** $X_s$ using $F$
7:         **update** $\boldsymbol{\Theta} \leftarrow$ update(step, $\nabla\ell(\boldsymbol{\Theta}, X_s)$)
8:     **end for**
9: **until** Convergence or $maxIter$ is reached

---

### 3.1 THEORETICAL FOUNDATIONS OF THE ALGORITHM

We first describe Problem 2, a multistep collocation of Problem 1, before reformulating and relaxing it to obtain the loss $\ell$ minimized by gradient descent, highlighting the connections with surrogate methods, adjoint methods, and shooting methods. As common for collocation, see Garg et al. (2010), we approximate the state by continuous piecewise polynomials of degree $K$ on a subdivision of $[0,T]$: $0=t_1<\ldots<t_N=T$. To harmonize polynomial representations and later optimize performance by precomputing matrices, we rescale time within each subinterval to [0,1] using the affine change of variable: $t=t_i+\tau h_i$ on the $i$th subinterval. We use the Lagrange basis $(l_j)_{j=0,\ldots,K}$ associated to $(\tau_j)_{j=0,\ldots,K}$, the LGR nodes of order $K$, and $\mathbf{x}_{ij}=\mathbf{x}(t_i+h_i\tau_j)$, the state values at collocation nodes. Each component of the state is an independent polynomial of time. We represent the state and its time derivative using two matrix-valued functions $\mathbf{V}(\tau)$ and $\mathbf{D}(\tau)$, see appendix D and the vector $\mathbf{X_i}=((\mathbf{x}_{ij_1})_{j\in[K]}^T, \ldots, (\mathbf{x}_{ij_n})_{j\in[K]}^T)^T$, obtained by stacking $\mathbf{x}_{ij}$ by component then index $j$:

$$\forall t \in [t_i, t_{i+1}], \tau = \frac{t - t_i}{h_i}, \ \mathbf{x}(t) = \sum_{j=0}^{K} l_j(\tau)\mathbf{x}_{ij} = \mathbf{V}(\tau)\mathbf{X}_i, \ \dot{\mathbf{x}}(t) = \frac{1}{h_i}\sum_{j=0}^{K} l'_j(\tau)\mathbf{x}_{ij} = \frac{1}{h_i}\mathbf{D}(\tau)\mathbf{X}_i.$$

By substitution in Problem (1), we obtain the classical collocation formulation, Problem (2):

$$\min_{\substack{\boldsymbol{\Theta}, \mathbf{x}_0 \\ (\mathbf{X}_i)_i}} \quad \frac{1}{M} \sum_{i=1}^{N-1} \sum_{\substack{m\in[M] \\ t_m\in[t_i,t_{i+1}]}} \|\mathbf{V}\left(\frac{t_m - t_i}{h_i}\right) \mathbf{X}_i - \mathbf{x}(t_m)\|^2$$

s.t.     $\mathbf{V}(0)\mathbf{X}_1 = \mathbf{x}_0$                                Initial condition,     (2a)

        $\mathbf{V}(0)\mathbf{X}_i = \mathbf{V}(1)\mathbf{X}_{i-1}$,   $i \in \{2, ..., N-1\}$,   Continuity between subintervals,  (2b)

        $\mathbf{D}(\tau_j)\mathbf{X}_i = h_i f(\mathbf{x}_{ij}, \boldsymbol{\Theta})$.   $\substack{i\in\{1,\ldots,N-1\}, \\ j\in\{1,\ldots,K\}}$.   Dynamic at collocation nodes.  (2c)

By relaxing constraints (2b), the problem splits into independent subtrajectories with shared parameters $\boldsymbol{\Theta}$: we first focus on subintervals. On any subinterval $[a, a+h]$, we reorder constraints and reformulate Problem (2) as Problem (3), where constraints have a block diagonal invertible structure, repeating $n$ times a matrix $\tilde{\mathbf{D}}_K$ that only depends on $K$, see details in appendices D, E. The right-hand side of the constraints involves a function $\mathbf{F}$ that stacks evaluations of $f$ at collocation nodes. The terms associated to the the $k$th component of the state are $(\mathbf{X}_{k(K+1)}, hf(\mathbf{x}(\tau_1), \boldsymbol{\Theta})_k, \ldots, hf(\mathbf{x}(\tau_K), \boldsymbol{\Theta})_k)^T$. Upon inverting $\tilde{\mathbf{D}}$, we obtain Problem (4), equivalent to Problem (2) on a single subinterval:

$$\min_{\substack{\boldsymbol{\Theta}, \\ \mathbf{X}}} \quad \frac{1}{M} \sum_{m=1}^{M} \|\mathbf{V}(\frac{t_m - a}{h})\mathbf{X} - \mathbf{x}(t_m)\|^2 \quad (3) \qquad \min_{\substack{\boldsymbol{\Theta}, \\ \mathbf{X}}} \quad \frac{1}{M} \sum_{m=1}^{M} \|\mathbf{V}(\frac{t_m - a}{h})\mathbf{X} - \mathbf{x}(t_m)\|^2 \quad (4)$$

s.t.   $\tilde{\mathbf{D}}\mathbf{X} = \mathbf{F}(\mathbf{X}, \boldsymbol{\Theta})$.                        s.t.   $\mathbf{X} = \tilde{\mathbf{D}}^{-1}\mathbf{F}(\mathbf{X}, \boldsymbol{\Theta})$.

Appendix E shows that the first column of $\tilde{\mathbf{D}}_K^{-1}$ is all ones, and the first row is all zeros but the first element. Denoting by $\mathbf{D}_K^{-1}$, the first principal minor of $\tilde{\mathbf{D}}_K^{-1}$, and $\tilde{\mathbf{F}}(\mathbf{X}, \boldsymbol{\Theta})_k = (f(\mathbf{x}(\tau_1), \boldsymbol{\Theta})_k, \ldots, f(\mathbf{x}(\tau_K), \boldsymbol{\Theta})_k)^T$, we recover equation 5, the Gaussian quadrature from the

LGR collocation see Garg et al. (2011a;b): $\forall k \in \{1, \ldots, n\}, \forall j \in \{1, \ldots, K\}$,

$$\tilde{\mathbf{D}}_K^{-1}\mathbf{F}(\mathbf{X}, \boldsymbol{\Theta})_k - \mathbf{X}_{jk} = \mathbf{X}_{k(K+1)} + h\mathbf{D}_K^{-1}\tilde{\mathbf{F}}(\mathbf{X}, \boldsymbol{\Theta})_k \approx \int_0^{h\tau_j} f(\mathbf{x}(t), \boldsymbol{\Theta})_k - \dot{\mathbf{x}}_k(t)dt \quad (5)$$

We relax the continuity constraint between subintervals, and the constraints of Problem (4) that are promoted through a quadratic penalty weighted by $\rho > 0$, as in Augmented Lagrangian relaxations:

$$\min_{\substack{\boldsymbol{\Theta}, \\ (\mathbf{X}_i)_i}} \frac{1}{M} \sum_{i=1}^{N-1} \underbrace{\sum_{\substack{m=1,\ldots,M \\ t_m \in [t_i, t_{i+1}]}} \left\| \mathbf{V}\left(\frac{t_m - t_i}{h_i}\right) \mathbf{X}_i - \mathbf{x}(t_m) \right\|^2}_{\substack{r_i(\mathbf{X}_i) n \text{ 1 Dimensional least square regression} \\ \text{estimating the values at LGR nodes from data}}} + \rho \underbrace{\sum_{i=1}^{N-1} \|\tilde{\mathbf{D}}^{-1}\mathbf{F}(\mathbf{X_i}, \boldsymbol{\Theta}) - \mathbf{X_i}\|^2}_{\ell(\boldsymbol{\Theta}, \mathbf{X}) \text{ system inversion}} \quad (6)$$

## 3.2 Resolution method and qualitative discussion

Problem (6) is solved by alternate descent with Algorithm 1. Estimations of the trajectory at LGR nodes using linear regressions ($r_{il}$ terms / L. 6 of the pseudo-code) alternate with gradient descent on system inversion problem ($\ell(\boldsymbol{\Theta}, \mathbf{X})$ / L. 7 of the pseudo-code). Solving Problem (3) directly suffers in practice from the same issues detailed in Roesch et al. (2021): gradients are relevant only when the integrated trajectory is close to data, although numerical integration remains computationally costly, as evidenced in Figure 1 by the initial plateau of ODE Solver methods from a random initialization. Allowing numerical integration error and computing the gradient along the interpolation avoids these issues. We set $\rho=1$ as this scales gradients that are rescaled after by Adam, Kingma and Ba (2014).

**Denoising, filtering and connection to other methods** Noise on data translates into interpolation error which is exacerbated by the non-uniform distribution of the LGR nodes over the interval $[0,1]$ (see Figure (2)). To mitigate this, we employ a denoising Savistzky-Golay filter Savitzky and Golay (1964), detailed in the Appendix A, although more advanced approaches can be beneficial (L. 2 of Algorithm (1)). An alternative is, in later iterations, to retain state values at LGR nodes as decision variables and alternate descents, as in the extension to unobserved components in Section 5 and Niu et al. (2016). As collocation methods are implicit Runge-Kutta methods, this approach reduces to a batched version of BPTT, with a connection to adjoint methods as our choice of collocation is symplectic, the upside being that numerical integrations are updated by gradient descent.

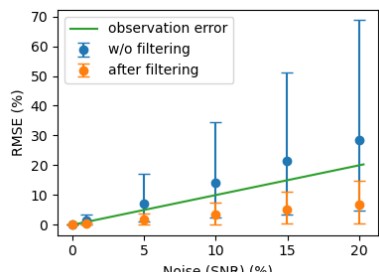

Figure 2: RMSE of the values at collocation nodes: irregular sampling amplifies noise (blue), filtering recovers accuracy. Sampling at $\Delta t = 0.01$.

**Continuity constraints between subintervals** although relaxed can be recovered by using overlapping intervals sharing the same data. When successive interpolations coincide on overlapping intervals, integrals along the interpolation can be computed in parallel across subintervals. Since ODEs are Markovian, an integrated trajectory that follows the interpolation will match the entire horizon recursively. Numerical error from collocation and discrepancies between interpolations may lead to the failure modes described in section 6. This is similar to shooting methods and checkpointing.

**Speed-ups:** While ODE solver methods are sequential, our approach is fully parallel. Furthermore, when parameters $\boldsymbol{\Theta}$ can be partitioned by state components -as in polynomial dynamics— Problem (6) can be decomposed by component: the algorithm scale linearly with the number of state dimensions and is hence well suited for high-dimensional systems.

## 3.3 Theoretical guarantees and asymptotic behavior

We consider a single interval $[a, a + h]$, with observations at collocation nodes $a + \tau_i h$. We denote by $\mathbf{x}$, the data-based polynomial interpolation. $f$ is assumed $L_f$-Lipschitz in state, and the Picard-Lindelöf theorem guarantees the ODE 1a solution's referred to as $\hat{\mathbf{x}}_{\boldsymbol{\Theta}}$, uniqueness for parameters $\boldsymbol{\Theta}$ and initial value $\hat{\mathbf{x}}_{\boldsymbol{\Theta}}(a) = \mathbf{x}(a)$. We denote by $L_1(\boldsymbol{\Theta})$, the loss of Problem 1 and by $L_2(\boldsymbol{\Theta})$ our loss. We prove two bounds linking our method to the original problem for noiseless and noisy settings.

### 3.3.1 Bounds at convergence and at a given iterate for noiseless observations

We prove two results, one asymptotic, and the other valid for any iterate $\boldsymbol{\Theta}$ of the Algorithm:

**Theorem 3.1.** *On a single subinterval, assuming a) noiseless estimation of the dynamic at collocation nodes, b) the function $f$ from 1a, Lipschitz wrt to the state.*

*i) should the descent find the optimum $\boldsymbol{\Theta}^*$ of Problem 6, such that $L_2(\boldsymbol{\Theta}^*)=0$ then*

$$L_1(\boldsymbol{\Theta}^*) \leq K(Ch^{2K-1})^2 + K(L_f h C_K h^{2K-1})^2 \tag{7}$$

*ii) For any $\boldsymbol{\Theta}$, of value $L_2(\boldsymbol{\Theta})=\delta h^2$, $\exists L_g>0, C>0$ involved in $D(h,K,\delta)=\frac{\sqrt{\delta}+Ch^{2K-2}}{\tau_K}+L_g\tau_K h$:*

$$L_1(\boldsymbol{\Theta}) \leq K(Ch^{2K-1})^2 + L_2(\boldsymbol{\Theta}) + K(L_f D(h,K,\delta)he^{L_f h})^2 \tag{8}$$

*Proof.* By definition, for noiseless observations, the loss of Problem 1 is $L_1(\boldsymbol{\Theta})$, our loss is $L_2(\boldsymbol{\Theta})$:

$$L_1(\boldsymbol{\Theta})=\sum_{i=1}^{K}\|\mathbf{x}(a+h\tau_i)-\hat{\mathbf{x}}_{\boldsymbol{\Theta}}(a+h\tau_i)\|^2=\sum_{i=1}^{K}\|\int_a^{a+h\tau_i}\dot{\mathbf{x}}(s)-f(\hat{\mathbf{x}}_{\boldsymbol{\Theta}}(s),\boldsymbol{\Theta})ds\|^2 \tag{9}$$

$$L_2(\boldsymbol{\Theta})=\sum_{i=1}^{K}\|\int_a^{a+h\tau_i}\dot{\mathbf{x}}(s)-f(\mathbf{x}(s),\boldsymbol{\Theta})ds\|^2 + o(h^{2K-1}) \quad \text{with Truncature error} \tag{10}$$

First, $f$'s Lipschitzness, implies the bound 12, constituted of two terms studied afterwards.

$$\|L_1(\boldsymbol{\Theta}) - L_2(\boldsymbol{\Theta})\| \leq \sum_{i=1}^{K}\|\int_a^{a+h\tau_i}f(\hat{\mathbf{x}}_{\boldsymbol{\Theta}}(s),\boldsymbol{\Theta})ds - f(\mathbf{x}(s),\boldsymbol{\Theta})ds\|^2 + K(C_g h^{2K-1})^2 \tag{11}$$

$$\leq \underbrace{K(C_g h^{2K-1})^2}_{\substack{\text{Collocation error,}\\\text{controlled by } h \text{ and } K \text{ (see Figure 9 and appendix H)}}} + \underbrace{K(L_f\|\mathbf{x}-\hat{\mathbf{x}}_{\boldsymbol{\Theta}}\|)^2)}_{\text{Solution-Interpolation distance}} \tag{12}$$

i) $L_2(\boldsymbol{\Theta}^*)=0$ implies that $\boldsymbol{\Theta}^*$ is the collocation (2)'s optimum, and truncature error gives 7.

ii) Introducing $e(t)=\mathbf{x}(t)-\hat{\mathbf{x}}_{\boldsymbol{\Theta}}(t)$, $f$'s Lipschitzness and the definition of $\hat{\mathbf{x}}_{\boldsymbol{\Theta}}$ give:

$$\forall t \in [a, a+h\tau_K], \|e(t)\| \leq \int_a^t \underbrace{\|\dot{\mathbf{x}}(s) - f(\mathbf{x}(s),\boldsymbol{\Theta})\|}_{\leq D(h,K,\delta),\text{see Appendix F}} + L_f\|e(s)\|ds$$

Gronwall's lemma implies, $\forall t\in[a,a+h\tau_K], \|e(t)\|\leq D(h,K,\delta)(t-a)e^{L_f(t-a)}\leq D(h,K,\delta)he^{L_f h}$

Which substituted into 10 concludes the proof of 8. $\qquad\square$

Error grows exponentially with interval length and $L_f$, which is fatal given chaotic systems can be learned. Bounds 7 and 8 offer guarantees that are usually missing for gradient matching and show the relevance of solutions when data allows an accurate interpolation.

### 3.3.2 Bounds in the noisy setting

Our approach is affected by the error at collocation nodes, rather than the noise on data. We denote by $\tilde{\mathbf{x}}$ the interpolation with error at collocation nodes. The triangle inequality leads to a bound similar to 12, with a supplementary quadratic term in the norm of the error:$\|L_1(\boldsymbol{\Theta}) - L_2(\boldsymbol{\Theta})\| \leq K(C_g h^{2K-1})^2 + KL_f^2(\|\mathbf{x}-\hat{\mathbf{x}}_{\boldsymbol{\Theta}}\|^2 + \|\tilde{\mathbf{x}}-\mathbf{x}\|^2)$, and additional term to the function $D(h,K,\delta)$. This is however a worst case scenario on a subinterval, and pessimistic as the method tends to average the error on long horizons through multiple overlapping intervals, see for instance Figure 3.

## 4 Experiments

We benchmark time, accuracy, noise robustness, and model convergence on common canonical models of the system identification literature. Each experiment simulates dynamics from a random initial condition, then runs algorithms on observations with Gaussian noise. Results are averaged over multiple initial conditions and noise seeds. We initialize parameters by 0 for polynomial dynamics as in this case, the system inversion problem reduces to a linear regression and convergence is not

Table 2: RMSE of $\dot{\mathbf{x}}$(%) on polynomial dynamics: for $T$=40, $h$=1 and $K$=30 our method outperforms SINDy in noisy settings. We included results with a sequential thresholding combination to highlight the good performance of our method without sparsity. For the Rossler, even for 20% noise, with sparsity, we recover the true support and error on coefficients is below 1%.

| | | | | NOISE | |
|---|---|---|---|---|---|
| MODEL | METHOD | 0% | 5% | 10% | 20% |
| LORENZ63 | SINDY | 0.18 | $7.5 \pm 5.9$ | $10.77 \pm 0.4$ | $22.95 \pm 3.4$ |
| LORENZ63 | IMATCH | 0.25 | $1.6 \pm 1.4$ | $4.63 \pm 3.6$ | $8.81 \pm 4.3$ |
| LORENZ63 | IMATCH THRESH | **0.05** | $\mathbf{1.0 \pm 0.6}$ | $\mathbf{2.8 \pm 2.6}$ | $\mathbf{6.3 \pm 7.1}$ |
| ROSSLER | SINDY | 0.02 | $4.3 \pm 0.5$ | $12.15 \pm 1.0$ | $26.95 \pm 2.3$ |
| ROSSLER | IMATCH | $< 10^{-2}$ | $0.5 \pm 0.1$ | $0.92 \pm 0.2$ | $2.11 \pm 0.5$ |
| ROSSLER | IMATCH THRESH | $< 10^{-2}$ | $\mathbf{0.3 \pm 0.1}$ | $\mathbf{0.9 \pm 0.2}$ | $\mathbf{1.5 \pm 0.9}$ |
| DUFFING | SINDY | **0.01** | $5.6 \pm 0.3$ | $9.79 \pm 0.3$ | $13.27 \pm 1.2$ |
| DUFFING | IMATCH | 0.34 | $\mathbf{1.1 \pm 0.9}$ | $\mathbf{2.0 \pm 1.9}$ | $\mathbf{4.2 \pm 4.3}$ |

impacted by initialization. Otherwise, we use the same methods as in Neural ODEs examples. We compare our method with baselines that exploit the structure of $f$, like SINDy when $f$ is a linear combination of nonlinear terms. The algorithm is implemented using PyTorch and JAX and Julia, tested on an Intel Xeon Platinum 8260 48-core server. We examine the performance in Section 4.1, higher-dimensional problems in Section 4.2, failure modes in Section 6, and complexity in Appendix H.

### 4.1 RAW PERFORMANCE ON THE LEARNING FROM NOISY OBSERVATIONS

We first consider canonical examples of chaotic systems: the Lorenz 63 attractor Lorenz (1963), the Rossler attractor Rössler (1976), the Duffing model Duffing and Emde (1918). Those systems are of dimensions up to 4 and are polynomials of degree up to 3.

**Learning Polynomial dynamics:** for each system, we fit the coefficients of polynomial dynamics of degree 3 that contain the original equations along with other terms and compare our method (Integral Matching - IMATCH) to the SINDy method for different levels of noise. Results in Table 2 show our algorithm learns meaningful models and is more robust to noise than SINDy. The lack of regularization in our method may explain SINDy's edge in noiseless cases due to the model's sparsity.

Table 3: Comparison of runtime on the a spiral dynamic in Chen et al. (2018) using a neural network with one hidden layer and 50 neurons, results averaged over 40 runs. Our algorithm (IMATCH) was found to compute gradients almost two orders of magnitude faster than the backpropagation through the solver (BPTT) and three orders of magnitude faster than the adjoint. The default parameters from the official Neural ODE library were used. The results are given as the 1st and 9th decile intervals.

| METHOD | RMSE $\dot{\mathbf{x}}$ (%) | GRADIENT ESTIMATION TIME (S) | # NFES $(10^6)$ | TOTAL TIME (S) | SPEED-UP PER GRADIENT |
|---|---|---|---|---|---|
| IMATCH | 1.61 [1.47, 1.77] | $1.8 \ 10^{-5}$ | 3.1 | 2.08 [1.01, 3.69] | |
| BPTT | 1.78 [1.19, 3.37] | $5.0 \ 10^{-2}$ | 3.9 | 100.5 [88.9, 116.3] | 2777 |
| ADJOINT | 1.83 [1.15, 2.57] | $4.5 \ 10^{-1}$ | 28.8 | 959.8 [909., 993.9] | 25000 |

Table 4: Wall-clock time on CPU (Intel i7) (statistics over 20 runs), a 30x+ speed-up on the learning the Lorenz63 model with Neural Networks with 5% added noise, details in section 4. The closest contender BPTT was not able, in an hour to match the performance our method achieved in 3 minutes.

| METHOD | RMSE $\dot{\mathbf{x}}$ (%) 3 MIN. | RMSE $\dot{\mathbf{x}}$ (%) 5 MIN. | RMSE $\dot{\mathbf{x}}$ (%) 10 MIN. |
|---|---|---|---|
| IMATCH(OURS) | $4.31 \pm 0.5$ | $3.8 \pm 0.4$ | $3.4 \pm 0.4$ |
| BPTT | $41.51 \pm 3.29$ | $32.20 \pm 2.46$ | $21.83 \pm 2.13$ |
| ADJOINT | $91.87 \pm 2.93$ | $87.89 \pm 4.37$ | $80.73 \pm 4.79$ |

**Learning Neural ODES** We consider the same damped oscillator as in Chen et al. (2018) and a simple network with one hidden layer of size 50 (202 parameters) and ReLU activation, as shown in Table 3. On this task, our method outperformed ODE Solver-based approaches on wallclock time by nearly two orders of magnitude. In the detail, we report the global wall clock time, time to evaluate a gradient for a batch of 20 observations, and the number of function evaluations. In Figure 1, we present a comparison on a training trajectory with the same data and initialization strategy. The $x$ axis is the number of function evaluations. While the figure represents one training trajectory, the orders of magnitude are consistent across multiple experiments. In total, our method achieves similar accuracy on the test set with 95 to 320 times fewer network evaluations than ODE Solver methods. On these instances, gradient estimations are up to 25,000 times faster than adjoint methods due to batching and parallelism. We trained a ResNet architecture with 300 hidden units (90,000 parameters) and two shared residual blocks with ReLU activations, on the Lorenz63 model and present results in Table 4. On a Tesla T4 GPU, an RMSE of around 1% was obtained within 20 minutes with our method, while 3% was achieved on CPU within 15 minutes. See appendix B for more details.

**Learning coefficients in nonlinear structures** The FitzHugh–Nagumo model FitzHugh (1961) involves three parameters in a rational function. Our method achieves error under 2% in coefficients, matching the Bayesian approach in Calderhead et al. (2008) while being twice as fast.

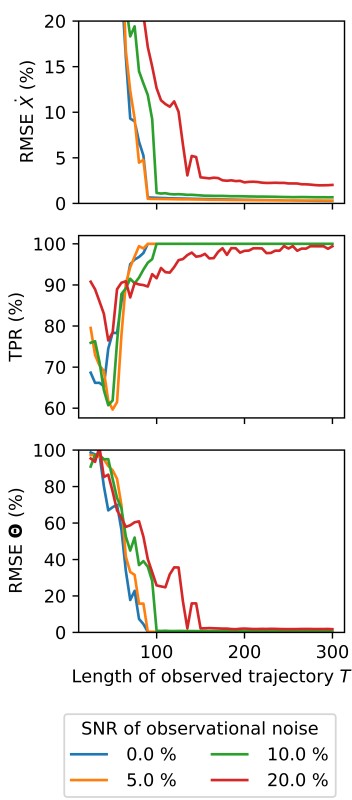

Figure 3: For the Lorenz96 model with $F = 16$, for various levels of noise and observed length, we plot (top) The RMSE of the error on the derivative, (middle) the true positive rate of the non zero terms identified in equations, (bottom) the relative error on parameters

### 4.2 Learning chaotic systems in high dimensions

The Lorenz (1996) models represent chaotic systems with states on a circle, influenced only by neighboring points. It is described by sparse polynomial ODEs with a forcing term $F$ whose value increases the chaoticity of the system: $\dot{x}_i = (x_{i+1} - x_{i-2})x_{i-1} - x_i + F, \ i = 1, \ldots, n$. The 40 dimension model with a forcing term $F = 16$ exhibits 9 positive eigenvalues for the linearized equation around the equilibrium, leading a very chaotic behavior in high dimensions, see Sapsis and Majda (2013). We consider trajectories initially disturbed from equilibrium by $10^{-16}$, below common tolerance used with ODE Solvers. We promoted sparsity using a simple sequential thresholding heuristic. Results in Figure 3 shows a phase transition which we believe is linked to sparsity and high dimensionality. The training on polynomial dynamics, including sequential thresholding runs in less than 30s for a 40 dimensions system, using threading to perform computations in parallel while considering the 861 monomial of degree up to 2. Appendix C presents a more comprehensive benchmark with a dynamic with $F = 32$ with even greater turbulence and chaos.

### 4.3 Conclusion

Benchmarks suggest the method's potential and noise robustness for nonlinear ODEs, with parameter nonlinearities under full system observability. We demonstrate its extension to partially observed systems and provide an experiment on the Lorenz attractor in 5.1.

## 5 Extension to partial observations

For practical applications of partially observed systems, only ODE solver methods are available, as gradient matching cannot estimate initial conditions and derivatives. A key challenge of ODE solver methods is the joint learning of the initial condition and the dynamics for latent dimensions, which is

addressed in Ayed et al. (2019); Lu et al. (2022) through the use of two different neural networks trained jointly via gradient descent. These networks use delayed observations as inputs, justified by Taken's theorem Takens (1981), which proves unobserved dimensions exist within the manifold of delayed observations in chaotic systems. Our method shows that in the case of the Lorenz63 attractor, the initial condition can estimated from past observations without regular sampling.

The algorithm (pseudo-code Algorithm 2 in Appendix G) solves the same problem 6, with the same loss but alternates between two steps: an evaluation of the trajectory at collocation nodes using a reconstruction method for unobserved components and a gradient descent to identify parameters of the system. By using the same loss function, the extended variant retains the theoretical properties and insights presented in Section 3, including the batching strategies, by subinterval and component. Compared to a full collocation approach, equivalent to applying BPTT to the implicit Runge-Kutta method from the collocation, we use interpolation to reduce the problem's size.

We rewrite the problem 6 on the subinterval $i$ and the component of the state $k$, by splitting the state variables into two classes: observed and unobserved components, splitting variables of the objective function and of the function $\mathbf{F}$ accordingly. At collocation nodes, observed components are estimated using data and regression $\hat{\mathbf{X}}_i$. Unobserved components are denoted by the decision variables $\tilde{\mathbf{X}}_i$. The objective function splits into terms associated to observed components $o_{ik}(\hat{\mathbf{X}}_i, \tilde{\mathbf{X}}_i, \mathbf{\Theta})$ for $k \leq p$, and terms relative to unobserved components $u_{ik}(\hat{\mathbf{X}}_i, \tilde{\mathbf{X}}_i, \mathbf{\Theta})$ for $k > p$. The relaxed problem is:

$$\min_{\substack{\mathbf{\Theta}, \\ (\mathbf{X}_i)_{i=1}^{N-1}}} \sum_{i=1}^{N-1} \{ \sum_{k=1}^{p} \underbrace{\|\mathbf{D_K}^{-1}\mathbf{F}(\hat{\mathbf{X}}_i, \tilde{\mathbf{X}}_i, \mathbf{\Theta})_k - \hat{\mathbf{X}}_{ik}\|^2}_{o_{ik}(\hat{\mathbf{X}}_i, \tilde{\mathbf{X}}_i, \mathbf{\Theta})} + \sum_{k=p+1}^{n} \underbrace{\|\mathbf{D_K}^{-1}\mathbf{F}(\hat{\mathbf{X}}_i, \tilde{\mathbf{X}}_i, \mathbf{\Theta})_k - \tilde{\mathbf{X}}_{ik}\|^2}_{u_{ik}(\hat{\mathbf{X}}_i, \tilde{\mathbf{X}}_i, \mathbf{\Theta})} \}$$

We utilized JAX to compute gradients, combining observed and decision state variables. Future reconstructions start by reusing past interval values from a subinterval pool. A version with the nonlinear solver Ipopt was implemented to incorporate equation constraints or penalties, such as energy conservation, into learning. In some cases, additional quadratic penalties help maintain continuity of unobserved components, though it wasn't needed for the chaotic system used afterward.

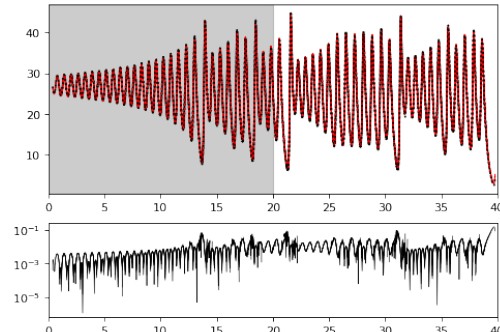

Figure 4: On the left, a scatter plot of the reconstructed component vs the ground truth. On the right, a scatter plot between the ground truth and the reconstructed model, once an affine change of variable has been found

## 5.1 NUMERICAL EXPERIMENTS

In this part, we experimented on polynomial dynamics of degree 2 for terms containing the observed component and 2 in the unobserved components. We consider the Lorenz oscillator for which we only observe the first two components. We approximate the dynamic on subintervals of length 1 by polynoms of degree $K = 30$ and are given observations over $[0, 20]$ sampled every $\Delta t = 0.01$.

When integrated over $[0, 20]$, the learnt model leads to a integration error over the three dimensions below $1\%$. The comparison of the reconstructed trajectory with the unobserved ground truth requires care as affine changes of variable leads to ODEs representing the same system. Figure 4 illustrates this phenomenon as the left plot represents the phase plot between the ground truth and the reconstructed trajectory, while the right plot shows the phase upon transformation with a suitable change of variable found by a linear regression of $R^2 = 0.99995$. Our method not only captured the attractor, it provided an accurate estimation of the unobserved component.

Figure 5: Top: Reconstructed trajectory (black) vs ground truth (red), Observation time (grey), (Bottom), relative error of the reconstruction: the learnt model captures the relevant dynamic with an average out of sample accuracy below $0.1\%$

Applying the affine change of variable, we are able to simulate the learnt system beyond the training horizon as in the figure 5. The grey area is the training horizon. The curve in red is the ground truth and in dotted black, the reconstructed and simulated unobserved component.

As curves are undistinguishable, the second plot below shows the relative error on the reconstructed trajectory, below 0.1% on average. Using sparse regression upon the change of variable, we recovered equations with less than $0.5\%$ error on the coefficients of the equations, remarkably, similar to the fully observed case.

## 6   LIMITATIONS AND FURTHER WORK

As mentioned in section (3.2), the accuracy of the interpolation is key for the performance of the method as it impacts two parts of the loss. The 1D, 1 subinterval system inversion term $\|\mathbf{D}^{-1}\mathbf{F}(\mathbf{X}, \mathbf{\Theta}) - (\mathbf{X} - \mathbf{X_0})\|^2$ is impacted in two ways by the interpolation error:

- In the numerical integration term along the trajectory via the $\mathbf{D}^{-1}F(\mathbf{X}, \mathbf{\Theta})$ terms, the non linearity $F$ can create a bias: the distribution of the error on $F(\mathbf{X} + \epsilon, \mathbf{\Theta})$ is not centered around $F(\mathbf{X}, \mathbf{\Theta})$, even if the approximation error $\epsilon$ is.
- In the difference $(\mathbf{X_i} - \mathbf{X_0})$, systematic bias is neutralized by the difference. As $\mathbf{X_0}$ is involved every integral of the subinterval, the variance on the initial condition is a key component for global accuracy and has motivated the Savitsky-Golay filter in Appendix (A). The use of multiple overlapping and random subintervals is a way to balance this variance on multiple points and subintervals, mitigating this single point of failure issue.

The impact of systematic bias, in the absence of filtering based on current equations, with the caveats on local optima mentioned in section (3.2), can often be mitigated by denoising with more data, and longer horizons. If not, our method can be used to initialize parameters prior to adjoint methods.

Another failure mode is about generalizability in the absence of prior structure on equations: in particular for Neural ODEs the trajectory may leave the manifold of the training data, and, in the absence of prior or additional penalty, the integrated trajectory becomes completely irrelevant. Oscillators such as the damped one in Chen et al. (2018) or Lokta-Volterra models Lotka (1926), as well as attractors have the characteristic that the system is trapped in a bounded manifold possibly and will visit regions multiple times over long horizons. In the case of the damped oscillator, as long as the trajectory remains in the envelope of interpolated trajectories, as solutions of ODEs do not intersect, the trajectory will generalize as it will cross a trained region.

The algorithm in this paper is based on fixed length subintervals and orders, but this is not a hard limitation. The integration order $K$ needs not be fixed (aside from easing implementation and providing a rigid computation graph), we can in principle use variable-length subintervals. Similarly as for SINDy methods, our method can be used to learn Partial Differential Equations using the method of lines to convert PDEs into a system of coupled ODEs.

## 7   CONCLUSION

We have studied the utilization of a particular collocation method for system identification of nonlinear dynamical systems, leveraging data to simplify the typically computationally intensive computations. Our efficient method requires fewer backpropagations to evaluate gradients at each step of the descent and introduces batching strategies, such as by subintervals and state components, to enable high parallelism and scalability linearly with dataset size, horizon length $T$, and system dimensions, contrasting with autoregressive ODE Solver methods.

**Note:**   If accepted, we intend to release codes of our method (Pytorch and JAX).

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

SUPPLEMENTARY MATERIALS

## A  DENOISING

As the LGR points are not uniformly distributed on the interval, being denser around boundaries, without denoising, the estimation error is significantly higher and has a higher variance than the noise level. However, in the center of the interval, noise is significantly lower than the noise level, dividing it by at least half, to more than a factor 4 with more data on experiments. Leveraging this observation,, we choose a window length and degree $K$, perform a linear regression to obtain denoised values at the window's center, and use a sliding window to estimate a denoised set of points along the trajectory. This larger set is used for the linear regression to estimate values at LGR point during the descent.

The experimental results, in Figure 2, consistently show that, without filtering, the RMSE of the estimation error at the LGR nodes is higher than the error in the observations. However, filtering significantly reduces the noise in the estimates, by approximately half to one-third compared to the original observations. This section illustrates the importance of denoising, rather than the study and analysis of this particular choice of method.

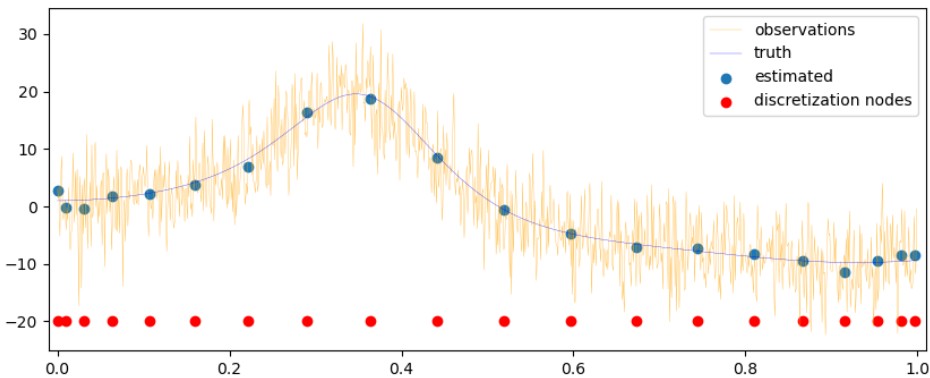

Figure 6: Observations corrupted by 40% noise. Without filtering, Estimates (blue) are irrelevant at boundaries due to uneven distribution of abscissae in the LGR nodes (red dots).

## B  TRAINING NEURAL ODEs USING INTEGRAL MATCHING

Contrary to library approaches such as in Brunton et al. (2016), Neural Networks bring less prior structure to the latent dynamic. As such, the true complexity of the manifold to learn and its translation as a data requirement especially the required length of observation to visit in different areas of the manifold is of utmost importance. Such characteristics are obviously problem specific, but there are connections with many areas studied in the physical context of finding either architectures that preserve physical quantities Raissi et al. (2017a;b) or promoting this through terms in the objective function. Promoting structure and respect of invariants brings structure to the parameters and reduces the complexity of the learning. All in all, our approach is perfectly compatible with such techniques, promoting invariants, and the loss function promotes the conservation of the Hamiltonian, though not enforcing it using projections as in Greydanus et al. (2019).

Similarly as for polynomial dynamics in section C, ie problems with more prior structure, a phase transition is observed and is linked to the architecture of the Network. Given a network with enough representative power to capture the dynamic, the phase transition is observed with regards to the availability of available data. There are several regimes, aside from terminal convergence to a relevant model that is observed in the following section where the algorithm is used to recover parameters of a dynamic within a class that contains the ground truth dynamic. We observed namely insufficient representative power and insufficient data to train the given architecture.

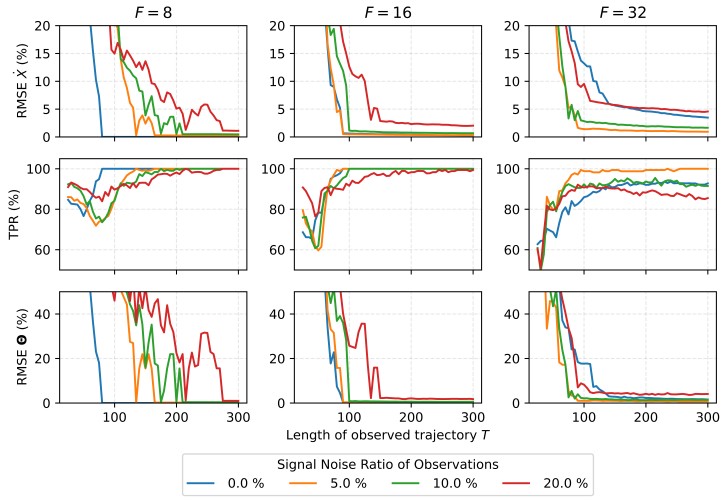

Figure 7: Phase transition: our algorithm converges to the ground truth model when given enough signal, consistently across different noise regimes. Columns: For increasing values of the forcing term $F$, corresponding to increasing chaoticity and difficulty, longer trajectories are required to recover the true dynamic. The first row indicates th RMSE of the time derivative, the second row, the true positive rate of nonzero terms recovered, the last row indicates the error on coefficients. The phase transition happens on the three metric, highlighting that past an amount of signal, our algorithm learnt the ground truth model. For $F = 8$, there is a phase transition, around $T = 80$ in the noiseless regime (blue) where the model is perfectly recovered. The greater the amount of noise, the later the phase transition. Interestingly, we observe that asymptotically on $T$, our model converges to a relevant solution for various noise levels, ie the red curve with the noise of $20\%$ converge to an error that keeps on decreasing with additional data. This is clearer in the second and third columns where the problem is more complex and the learning longer. After a phase transition, happening around $T = 100$, even $T = 130$ for $20\%$ noise, the performance keeps on improving. On $F = 32$, another surprising phenomenon appeared: on chaotic systems, mild levels of noise appear to be beneficial to help convergence. This is possibly linked to the sparsifying heuristic being suboptimal.

## C  FOCUS OF THE LORENZ 96

Contrary to the experiments in the core paper that contained no method to promote sparsity, aside from the implicit regularization of gradient descent and a small $\ell_1$ penalty that helped convergence, we used in these section a simple sequential thresholding heuristic: the problem was solved, then small values projected to $0$, then retrained on the subset of nonzero values, then thresholded again. Results are presented by Figure 7 for forcing terms $F = 8$, $F = 16$ and $F = 32$. We also provide a specific focus on $F = 8$ in Figure 8.

Better methods have been developed to recover sparse equations than the simple sequential heuristic, in a a sparse regression setting close to ours, but the fact that such a simple methods works well illustrates the interest of the loss and overall procedure. It should be noted, lastly, that such a computationally cheap sparsifying method is scalable to large dimensions. However, on large dimensions, for instance, $N = 200$, the number of monomials grows to more than 20,000 terms for a polynomial of degree 2, so that the method of postulating a library is doomed in higher dimensions. In such dimensions, the regression step to estimate the value at LGR nodes and the denoising process are no longer cheap, though easily treated in parallel.

However, using our method can provide an interesting option as the sequential thresholding can be used to filter and select a lower dimension set of features, so that the method speeds up as it converges to a sparser model. This points towards future work at the intersection of interpretability and sparsity on Neural ODEs.

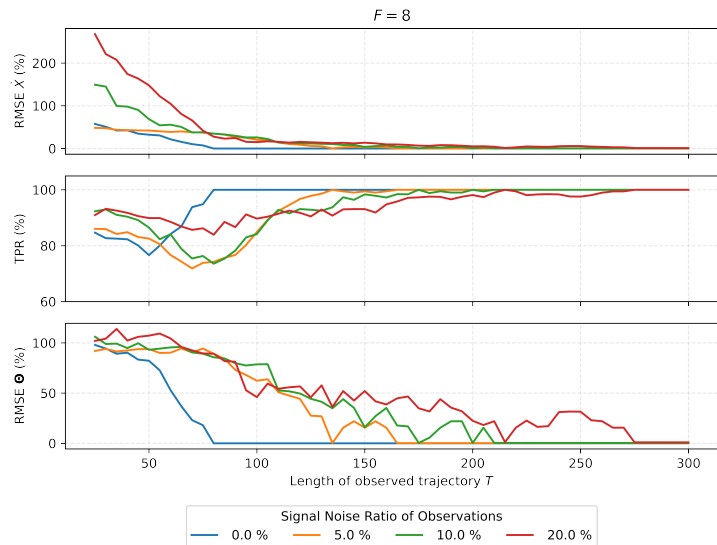

Figure 8: On the problem with $F = 8$, we present a more global view of the first column. The oscillations observed for the various metrics are likely explained by the suboptimality of the sparsifying heuristic and thresholding effects.

## D    REFORMULATION OF THE SINGLE SUBINTERVAL PROBLEM

The state polynomial interpolation and its time are written using matrix valued functions $\mathbf{V}$ and $\mathbf{D}$ on the subintervals rescaled to $[0, 1]$ and the vector obtained by stacking the values of the state at collocation nodes. Namely:

$$\forall t \in [t_i, t_{i+1}], \tau = \frac{t - t_i}{h_i}, \ \mathbf{x}(t) = \sum_{j=0}^{K} l_j(\tau) \mathbf{x}_{ij} = \mathbf{V}(\tau) \mathbf{X}_i, \ \dot{\mathbf{x}}(t) = \frac{1}{h_i} \sum_{j=0}^{K} l'_j(\tau) \mathbf{x}_{ij} = \frac{1}{h_i} \mathbf{D}(\tau) \mathbf{X}_i.$$

The formalism relies on the fact that each component of the state is a polynomial in time that is determined by the values of the state at collocation nodes only for the very same component. The problem is a 1D polynomial of time for each dimension and the Lagrange basis is the same for each dimension. Given the structure of $\mathbf{X}_i$, the functions $\mathbf{V}$ and $\mathbf{D}$ have a block diagonal structure (by component of the state), repeating $n$ times a matrix that is simply the decomposition of the polynomial within a Lagrange basis.

- $\mathbf{V}(\tau)$ repeats $n$ times $\mathbf{V_d}$ whose general term is $\mathbf{V}_{\mathbf{d}(i,j)}(\tau) = l_j(\tau)$,

- for $\mathbf{D}(\tau)$ repeats $n$ times $\mathbf{D_d}$ whose the general term $\mathbf{D}_{\mathbf{d}(i,j)}(\tau) = l'_j(\tau)$

Using these notations, the single subinterval problem has the following form:

$$\min_{\boldsymbol{\Theta}, \mathbf{X}} \quad \frac{1}{M} \sum_{m \in [M]} \left\| \mathbf{V}\left(\frac{t_m - a}{h}\right) \mathbf{X} - \mathbf{x}(t_m) \right\|^2$$

$$\text{s.t.} \qquad \mathbf{x}_0 = \mathbf{x}_0, \tag{13a}$$

$$\mathbf{D}(\tau_j)\mathbf{x} = h f(\mathbf{x}_j, \boldsymbol{\Theta}) \quad j \in 1...K. \tag{13b}$$

We reorder the constraints by components of the state first then time. By design of the collocation method, each dimension is interpolated separately. As such, grouping terms of each dimension separately, constraints naturally separate by dimension: introducing a $(K + 1) \times (K + 1)$ matrix $\mathbf{D}_K$, the constraints on the $k$th component of the state are:

$$\underbrace{\begin{pmatrix} 1 & 0 & ... & 0 \\ \dot{l}_0(\tau_1) & \dot{l}_1(\tau_1) & ... & \dot{l}_K(\tau_1) \\ \vdots & \vdots & \ddots & \vdots \\ \dot{l}_0(\tau_K) & \dot{l}_1(\tau_K) & ... & \dot{l}_K(\tau_K) \end{pmatrix}}_{\mathbf{D}_K} \cdot \underbrace{\begin{pmatrix} X_{k(K+1)} \\ X_{k(K+1)+1} \\ \vdots \\ X_{k(K+1)} \end{pmatrix}}_{X[kK:k(K+1)]}$$

$$= \underbrace{\begin{pmatrix} X_{k(K+1)} \\ hf(\mathbf{x}_1, \boldsymbol{\Theta})_k \\ \vdots \\ hf(\mathbf{x}_K, \boldsymbol{\Theta})_k \end{pmatrix}}_{F(\mathbf{X},\boldsymbol{\Theta})_k}$$

Where $\mathbf{X}[k(K+1):(k+1)(K+1)]$ is the projection of $\mathbf{X}$ on the span of $\{\mathbf{e}_{k(K+1)}, \ldots \mathbf{e}_{(k+1)(K+1)}\}$. As the collocation is of the same order for each dimension, the matrix $D_K$ which only depends on the collocation order is the same for each dimension, so that, stacking back every component, the matrix of constraints is block diagonal:

$$\underbrace{\begin{pmatrix} \mathbf{D}_K & & & \\ & \mathbf{D}_K & & \\ & & \ddots & \\ & & & \mathbf{D}_K \end{pmatrix}}_{\tilde{\mathbf{D}}} \cdot \mathbf{X} = \underbrace{\begin{pmatrix} F(\mathbf{X}, \boldsymbol{\Theta})_1 \\ F(\mathbf{X}, \boldsymbol{\Theta})_2 \\ \vdots \\ F(\mathbf{X}, \boldsymbol{\Theta})_n \end{pmatrix}}_{F(\mathbf{X},\boldsymbol{\Theta})}$$

One final observation (proof in the following section) is that the structure of the first row of $\mathbf{D}_K$ and the collocation structure imply that the first column of the inverse of $\mathbf{D}_K$ is only composed of 1s. Namely, we have:

$$\mathbf{D}_K^{-1} = \begin{pmatrix} 1 & 0 & ... & 0 \\ \vdots & & \hat{\mathbf{D}} & \\ 1 & & & \end{pmatrix}$$

This enables to compute the loss by multiplying by a $K \times K$ submatrix of $\mathbf{D}_K^{-1}$ rather than by a $(K+1) \times (K+1)$ matrix. For $K = 30$, this simple observations enables to reduce the number of operations to evaluate the product by 9% using Strassen ($O(K^{2.8})$). In the end, compared to a naive implementation that would consider a product with matrix of constraint of dimension $n(K+1) \times n(K+1)$, we have transformed the problem into $n$ products with matrices of dimension $K \times K$. As the matrix is fixed, it seems from experiments that the compilation performed in JAX is able to the product for further speedups.

## E   PROOF OF THE INVERTIBILITY OF DIFFERENTIATION MATRICES

Any element of the kernel of $\mathbf{D}_K$ can be interpreted a polynomial $P$ of degree $K$ represented in the LGR Lagrange basis. The last $K$ rows of $\mathbf{D}_K$ imply that $P$ is constant: the derivative of $P$ is a polynomial of degree $K - 1$ null at $K$ distinct points, hence null everywhere. The first row of the matrix $\mathbf{D}_K$ implies that this constant is null, ie $P = 0$. Subsequently, $\mathbf{D}$ is also invertible from its block diagonal structure of matrices $\mathbf{D}_K$. $\square$.

The first column $\mathbf{v}$ of matrix $\mathbf{D}_K^{-1}$ is a vector of ones: $\mathbf{v} = \mathbf{1}$. The first component is trivial. For the other ones, we use the adjoint matrix, algebraic manipulations and the interpretation as a

differentiation matrix. Namely, have that

$$\det(\mathbf{D}_K) = \det \begin{pmatrix} 1 & 0 & ... & 0 \\ \dot{l}_0(\tau_1) & \dot{l}_1(\tau_1) & ... & \dot{l}_K(\tau_1) \\ \vdots & \vdots & \ddots & \vdots \\ \dot{l}_0(\tau_K) & \dot{l}_1(\tau_K) & ... & \dot{l}_K(\tau_K) \end{pmatrix}$$

for any $k \neq 0$

$$\det(\mathbf{D}_K) = \det \begin{pmatrix} \dot{l}_1(\tau_1) & ... & \dot{l}_k(\tau_1) & ... & \dot{l}_K(\tau_1) \\ \vdots & \vdots & \vdots & \vdots & \vdots \\ \dot{l}_1(\tau_K) & ... & \dot{l}_k(\tau_K) & ... & \dot{l}_K(\tau_K) \end{pmatrix}$$

Using the adjoint matrix formula for the inverse of $\mathbf{D}_K$), to prove that $\mathbf{v} = \mathbf{1}$, we need to show that the first row of the adjoint matrix, ie the cofactors are each equal to the the determinant of $\det(\mathbf{D}_K)$. Namely, we need to show that, for any $k \neq 0$, $\det(A_k) = \det(\mathbf{D}_K)$ where:

$$\det(A_k) = (-1)^k \det \begin{pmatrix} \dot{l}_0(\tau_1) & ... & \dot{l}_{k-1}(\tau_1) & \dot{l}_{k+1}(\tau_1) & ... & \dot{l}_K(\tau_1) \\ \vdots & \vdots & \vdots & \vdots & \vdots & \vdots \\ \dot{l}_0(\tau_K) & ... & \dot{l}_{k-1}(\tau_K) & \dot{l}_{k+1}(\tau_K) & ... & \dot{l}_K(\tau_K) \end{pmatrix}$$

We form the difference $\Delta_k = \det(\mathbf{D}_K) - \det(A_k)$ and expand the determinant of $\mathbf{D}_K$ along the $k$th column, expand the determinant of $\mathbf{A}_k$ along the first column. The expansion exhibits the same minors obtained by removing the 0th and $k$th columns. We denote $\mu_{ik}$ the determinant of the minor obtained by removing the first and $i$th row of $\mathbf{D}_K$ and the first column and $k$th column of $\mathbf{D}_K$:

$$\Delta_k = \det(\mathbf{D}_K) - \det(A_k) = \sum_{i=1}^{K} (-1)^{i+k} \dot{l}_k(\tau_i) \mu_{ik} - \sum_{i=1}^{K} (-1)^k (-1)^{i+1} \dot{l}_0(\tau_i) \mu_{ik}$$

$$= \sum_{i=1}^{K} (-1)^{i+k} (\dot{l}_k(\tau_i) + \dot{l}_0(\tau_i)) \mu_{ik}$$

That is the difference is the determinant of a matrix $\mathbf{B}_k$

$$\Delta_k = \det(\mathbf{B}_k) = \det \begin{pmatrix} 1 & 0 & ... & 0 & ... & 0 \\ \dot{l}_0(\tau_1) & \dot{l}_1(\tau_1) & ... & \dot{l}_k(\tau_1) + \dot{l}_0(\tau_1) & ... & \dot{l}_K(\tau_1) \\ \vdots & \vdots & & \vdots & & \\ \dot{l}_0(\tau_K) & \dot{l}_1(\tau_K) & ... & \dot{l}_k(\tau_1) + \dot{l}_0(\tau_1) & ... & \dot{l}_K(\tau_K) \end{pmatrix}$$

Subtracting the first column from the $k$th column does not change the determinant but gives a new matrix $\tilde{\mathbf{B}}_k$ which is the same as the original matrix $\mathbf{D}_K$ except for the term on the first row and the $k$th column which equals $-1$. This matrix is not invertible: using the same interpretation as polynomials used earlier in this section, we deduce that constant polynomials ie vectors of $\mathbb{R}^{K+1}$ that are collinear to $v$ are in the kernel of this matrix.

The last $K$ rows of $\tilde{\mathbf{B}}_k v$ imply that the derivative of a constant polynomial is 0. The first row also evaluates to 0 so that $v \neq 0 \in \ker \tilde{\mathbf{B}}_k$.

Thus, $\forall k, \Delta_k = \det(\mathbf{B}_k) = 0$. $\square$.

## F PROOF OF THEORETICAL BOUNDS

For convenience, we recall the original loss $L_1$ and the surrogate loss $L_2$:

$$L_1(\boldsymbol{\Theta}) = \sum_{i=1}^{K} \|\mathbf{x}(a+h\tau_i) - \hat{\mathbf{x}}_{\boldsymbol{\Theta}}(a+h\tau_i)\|^2 = \sum_{i=1}^{K} \|\int_{a}^{a+h\tau_i} \dot{\mathbf{x}}(s) - f(\hat{\mathbf{x}}_{\boldsymbol{\Theta}}(s), \boldsymbol{\Theta}) ds\|^2 \quad (14)$$

$$L_2(\boldsymbol{\Theta}) = \sum_{i=1}^{K} \|\int_{a}^{a+h\tau_i} \dot{\mathbf{x}}(s) - f(\mathbf{x}(s), \boldsymbol{\Theta}) ds\|^2 + o(h^{2K-1}) \quad (15)$$

Given an iterate $\boldsymbol{\Theta}$, we denote $\delta = 1/h^2 L_2(\boldsymbol{\Theta})$ and have the following bound on $e(t) = \mathbf{x}(t) - \hat{\mathbf{x}}_{\boldsymbol{\Theta}}(t)$:

$$\forall t \in [a, a+h\tau_K], \|e(t)\| \leq D(h,K)(t-a)e^{L_f(t-a)} \leq D(h,K)he^{L_f h} \qquad (16)$$

where $D(h,K) = \frac{\sqrt{\delta} + Ch^{2K-2}}{\tau_K} + L_g \tau_K h$

*Proof* : using the definition of $\hat{\mathbf{x}}_{\boldsymbol{\Theta}}$:

$$\forall t \in [a, a+h], e(t) = \int_a^t \dot{e}(s)ds = \int_a^t \dot{\mathbf{x}}(s) - f(\hat{\mathbf{x}}_{\boldsymbol{\Theta}}, \boldsymbol{\Theta})ds \qquad (17)$$

$f$'s $L_f$ Lipschiztness and the triangle inequality then gives:

$$\|e(t)\| \leq \int_a^t \|\dot{\mathbf{x}}(s) - f(\mathbf{x}(s), \boldsymbol{\Theta})\| + L_f \|e(s)\|ds \qquad (18)$$

We define $\forall b \in [a, a+\tau_K h], g(b) = \dot{\mathbf{x}}(b) - f(\mathbf{x}(b), \boldsymbol{\Theta})$. $g$ is Lipschitz in time: for any $b$:

$$\| \int_a^{a+\tau_K h} (g(s) - g(b))ds \| \leq L_g (\tau_K h)^2$$

Hence, using the triangle inequality:

$$\|g(b)\| \tau_K h \leq \| \int_a^{a+\tau_K h} g(s)ds \| + L_g (\tau_K h)^2$$

By definition of $L_2$: $\| \int_a^{a+\tau_K h} g(s)ds \| \leq \sqrt{\delta h^2} + Ch^{2K-1}$.

Hence, $\forall b \in [a, a+\tau_K h], \|g(b)\| \leq D(h,K)$, where $D(h,K) = \frac{\sqrt{\delta} + Ch^{2K-2}}{\tau_K} + L_g \tau_K h$

Combining this bound with 16 and applying Grönwall's Lemma, we obtain the desired bound on the error. $\square$

## G    PSEUDO-CODE OF THE PARTIALLY OBSERVED ALGORITHM

---

**Algorithm 2** Reconstructed Subtrajectory Gradient Descent

---

**Input:** data $(t_m, x(t_m))_{m=1...M}$, approximation degree $K$, subinterval length $h$, state dimension $n$, observed dimension $p$, gradient update method
Build set of filtered points $F = \{(t_f, x_f(t_f)\}$ from data
**initialize** $\Theta$
**repeat**
  **Generate** a random set $S$ of subintervals of length $h$
  **for** $s$ in $S \cup \{s_0\}$ **do**
    **compute** $X_s \in \mathbb{R}^{p(K+1)}$ by solving $p$ Linear regressions on $s$, using data from $F$
    **compute** $\hat{X}_s = \gamma(\boldsymbol{\Theta}, X_s) \in \mathbb{R}^{(n-p)(K+1)}$ by solving the reconstruction problem
  **end for**
  **for** $k = 1$ to $n_{steps}$ **do**
    **initialize** Gradient $\nabla \ell_\Theta = 0$
    **initialize** Gradient $\nabla \ell_{\hat{X}_s} = 0$ for every $s \in S$
    **for** $s$ in $S$ **do**
      **compute** $\nabla_\Theta g(\hat{X}_s, \Theta, X_s)$
      **accumulate:** $\nabla \ell_\Theta += \nabla_\Theta g(\hat{X}_s, \Theta, X_s)$
    **end for**
    **update** $\Theta$: $\Theta \leftarrow$ update(step, $\nabla \ell_\Theta$)
    **for** $s$ in $S$ **do**
      **compute** $\nabla = \nabla_{\hat{X}_s} g + \nabla_{\hat{X}_s} r(\hat{X}_s, (\hat{X}_m)_{m \in S})$
      **accumulate:** $\nabla \ell_{\hat{X}_s} += \nabla X$
    **end for**
    **update** $\hat{X}_s$: $\hat{X}_s \leftarrow$ update(step, $\nabla \ell_{\hat{X}_s}$)
  **end for**
**until** $maxIter$ is reached

---

Table 5: Number of Evaluations for Different Integration Error Tolerances and $T = 1$. Our method requires at least one order of magnitude fewer evaluations than common methods.

| | ABS. TOLERANCE | | |
| METHOD | $10^{-3}$ | $10^{-6}$ | $10^{-8}$ |
|---|---|---|---|
| DORMAND PRINCE | $100 \pm 11$ | $199 \pm 28$ | $336 \pm 42$ |
| RADAU 5 | $148 \pm 26$ | $678 \pm 119$ | $2104 \pm 385$ |
| BDF | $95 \pm 16$ | $274 \pm 62$ | $649 \pm 113$ |
| LGR, $K = 5$ | 27 | 142 | 333 |
| LGR, $K = 8$ | 22 | 72 | 133 |
| LGR, $K = 20$ | 10 | 50 | 71 |
| LGR, $K = 30$ | 6 | 50 | 59 |

## H COMPARISON OF THE NUMBER OF BACKPROPAGATIONS WITH ODE SOLVERS METHODS

We complement the theoretical estimates from Table 1 by a experimental comparison on the Lorenz63 model. While adaptive methods involve varying step numbers during descent, our analysis provides a static estimation that hints at the significant computational gap between methods. To avoid interference with measurement times, we only compute the state at $t = T$ ($M = 1$). We present averaged results for ODE solvers recommended for Neural ODEs in Table 5. Experiments reveal that our method requires between 10 to 40 times fewer backpropagations on the neural network $f$ than standard methods for Neural ODEs. For the 40-dimensional Lorenz96, function evaluations decrease by a factor of 20 when comparing $K = 30$ to BDF. Looking at Table 5 and Figure 9, a natural question is the choice of $K$. Increasing $K$ improves accuracy but also increases data requirements and computational costs due to the super-quadratic complexity of matrix multiplication (Strassen or Fawzi et al. (2022)). A higher $K$ might also capture more noise, putting an emphasis on denoising. In our experiments, using $K = 30$ and $h = 1$ yielded satisfactory results in terms of accuracy and runtime.

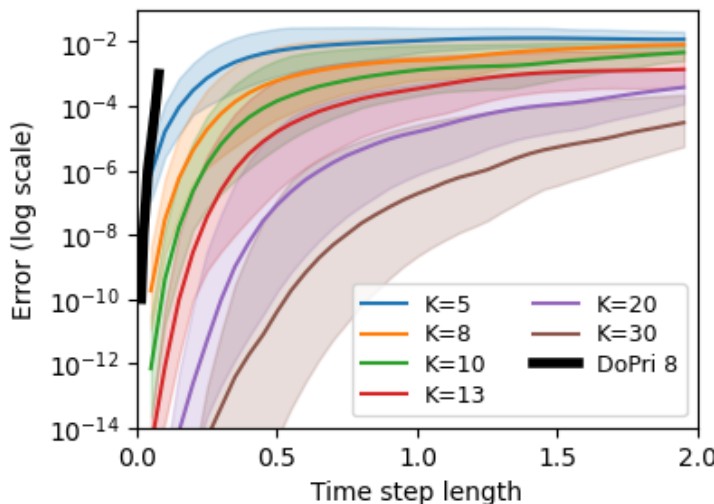

Figure 9: Integration error (avg.) on Lorenz63 vs. step length for DoPri and various orders.

