# OpenReview forum: "ODE Parameter Identification: An Integral Matching Approach"
_ICLR.cc/2025/Conference — Submitted to ICLR 2025_

### Official Review · Reviewer_Wjxv · 2024-11-01

**Soundness:** 2
**Presentation:** 2
**Contribution:** 3
**Rating:** 5
**Confidence:** 3

**Summary:**

This manuscript introduces a novel approach for parameter identification in nonlinear ODEs using integral matching. By leveraging collocation-based integral estimates, the authors present a method that demonstrates substantial computational efficiency and robustness to noise. The approach is benchmarked against traditional solver-based and surrogate methods, showing impressive speed improvements.

**Strengths:**

1.	The paper is well-motivated, with a comprehensive background and related work section that positions the proposed method within the existing literature.
2.	Experimental results indicate that the proposed method achieves significant speed improvements and demonstrates reduced sensitivity to noise, highlighting its practical advantages over existing methods.

**Weaknesses:**

1. Clarity in Theoretical Presentation: Section 3, which discusses the main theoretical results, would benefit from further clarification. Several aspects currently hinder readability:
- Inconsistent Notation: There is inconsistent notation for derivatives (e.g., $\dot{x}(t)$ vs. $l’(\tau)$), and mixed usage of symbols such as $\tau$, $\tau_j$, and $\tau_k$, which may confuse readers.
- Unexplained Symbols: Certain notations, such as $X_{il}$ and $X_{i,0}$, are introduced without adequate explanation, limiting clarity.
- Ambiguity in Notations: In Equation (6), two instances of $X_i$ appear. Clarification on whether they denote the same or distinct elements is advised.
- Change of Variables: The statement "the change of variable $t = t_i + \tau h_i$ rescales time to $[0,1]$" could be explained in greater detail to aid understanding.

These issues give the impression of insufficiently clear theoretical exposition. To improve readability and ensure clarity, I suggest that the authors thoroughly review and streamline their mathematical notations and formulations, providing explanations where necessary. Including a summary table of notations would be an effective way to guide readers.

2. Notation clarity in Tables:
- In Table 1, notation $\epsilon$ is not defined. Additionally, the arrows in the “Accuracy FWD/BWD” column lack explanation.
- In Table 3, entries in the "Gradient Estimation Time (s)" column, such as $510^{-2}$, appear confusing and could be rephrased for clarity.

**Questions:**

Could the authors clarify how RMSE is computed in this study? Specifically, what is the rationale for calculating RMSE based on $\dot{x}$ rather than $x$?

---

> ### Author Response · Authors · 2024-11-21
>
> We thank the reviewer for his careful review that helps enhance the clarity and quality of the manuscript. We have uploaded a revised manuscript where we have modified primarily section 3 (Pages 4,5,6), to clarify notations, link steps in the pseudo-code with the formulations and discussions.
>
> We have factored in the remarks of the reviewer to harmonize notation to simplify the reading and add more explanations.
> For instance, the statement relative"the change of variable " could be explained in greater detail to aid understanding.
> This is just an affine change of variable so that each polynomial of each subinterval is represented in the same basis which makes notations, and implementation easier as rescaled matrices can be precomputed once and for all. We have included a similar explanation in the revision.
>
> Notation clarity in Tables:
> 'In Table 1, $\epsilon$ is not defined.
> > We have fixed this, $\epsilon$ is the tolerance on numerical integration error.
> Additionally, the arrows in the “Accuracy FWD/BWD” column lack explanation.
> > Explanations are given in section 2 of the submission “Accuracy FWD/BWD''.
>
> Questions:
> > ''Could the authors clarify how RMSE is computed in this study? Specifically, what is the rationale for calculating RMSE on the derivative?''
>
> The RMSE of the derivative is computed along the noiseless exact trajectory. We reported it because most examples are chaotic systems so that the metric on the original state $x$ can be quickly irrelevant as systems might diverge quickly. We plan to add the RMSE on $x$ later this week to the relevant tables.

---

> > ### Comment · Reviewer_Wjxv · 2024-11-26
> > **Thank you for your response**
> >
> > Thank you for addressing my questions. I appreciate the effort you have put into clarifying notations and improving the explanations.
> >
> > However, in the revised version, the updated parts were not clearly marked, making it challenging to locate the changes. Additionally,  the experiments based on RMSE on the derivative $\dot{x}$ are not convincing, and I did not see the results based on RMSE on $x$ that you mentioned were planned for inclusion.
> >
> > For these reasons, I will maintain my original score. Thank you again for your response and revisions.

---

### Official Review · Reviewer_9Pjc · 2024-11-01

**Soundness:** 2
**Presentation:** 1
**Contribution:** 1
**Rating:** 3
**Confidence:** 3

**Summary:**

This paper introduces an integral matching strategy for system identification. However, there are several concerns regarding its formulation and soundness. The proposed model has not been sufficiently validated through experimentation. Addressing these weaknesses is essential for enhancing the credibility of the proposed model and improving its practical applicability in modeling complex dynamics.

**Strengths:**

The integral matching strategy appears promising because integration is a more mathematically stable and bounded operator than derivatives. This approach may lead to improved training stability and performance in modeling complex dynamics.

**Weaknesses:**

1.	A major concern arises from the separation of the dynamics into independent subintervals, which results in the loss of continuity between them after ignoring eq (2b). This separation could cause significant issues in the overall performance of the model since the connection between the subintervals is crucial. It is essential to conduct experiments to assess how well the model learns the relationships between these subintervals.
2.	The method could be improved by incorporating eq. (2b) in eqs (6) and (7), similar to how eq (2c) is treated. While this may not be computationally efficient, it is necessary to explore this modification experimentally to examine its impact on performance.
3.	The claim that eq (6) uses an Augmented Lagrangian is incorrect, as the method used is actually a penalty method. The Augmented Lagrangian requires solving a saddle system, but it generally provides better convergence to the minimizer of constrained problems compared to the penalty method. While the penalty method is conceptually simpler, it requires the penalty parameter $\rho$ to approach infinity for the constraints to be satisfied mathematically. Therefore, the choice of $\rho$ in the penalty method chosen by the authors is critical to ensuring constraint satisfaction. This parameter $\rho$ is a new hyperparameter not present in traditional NODEs, and a study of its effects is needed.
4.	The performance of the proposed method is highly dependent on the polynomial interpolation degree $K$. I recommend an ablation study to investigate the effects of varying $K$ to better understand its influence on model performance. In addition, comparisons with other node types beyond the LGR node leveraged in this study is also required.

**Questions:**

Please address the concerns mentioned in the weaknesses part above.

---

> ### Author Response · Authors · 2024-11-21
>
> We thank the reviewer for his careful review that helps enhance the clarity and quality of the manuscript. We have uploaded a revised manuscript where we have modified section 3 (Pages 4,5,6), to clarify notations, link steps in the pseudo-code with the formulations and discussions. We have added an appendix that explicits the link with ADMM.
>
> To answer the reviewer's specific questions:
> "1 A major concern arises from the separation of the dynamics into independent subintervals, which results in the loss of continuity between them after ignoring eq (2b). This separation could cause significant issues in the overall performance of the model since the connection between the subintervals is crucial. It is essential to conduct experiments to assess how well the model learns the relationships between these subintervals."
>
> > For fully observed systems, as discussed in the original manuscript and in the revised version in section 3.2, the relaxation of continuity in the collocated formulation is recovered by the interpolation and overlapping intervals as ODEs are Markovian. The continuity is implicitly contained in the interpolation.
>
> > "2 The method could be improved by incorporating eq. (2b) in eqs (6) and (7), similar to how eq (2c) is treated. While this may not be computationally efficient, it is necessary to explore this modification experimentally to examine its impact on performance."
>
> This is an interesting point that has proven  beneficial in instances of partially observed systems. For fully observed, no benefit as per above. We have added elements of this discussion in the revised version.
>
> > 3- "The claim that eq (6) uses an Augmented Lagrangian is incorrect, as the method used is actually a penalty method. The Augmented Lagrangian requires solving a saddle system, but it generally provides better convergence to the minimizer of constrained problems compared to the penalty method. While the penalty method is conceptually simpler, it requires the penalty parameter
>  to approach infinity for the constraints to be satisfied mathematically. Therefore, the choice of in the penalty method chosen by the authors is critical to ensuring constraint satisfaction. This parameter is a new hyperparameter not present in traditional NODEs, and a study of its effects is needed."
>
> We agree this is a penalty method. However, due to the separation of problems this penalty parameter does not affect the result as it scales gradient of the system inversion problem that are rescaled anyway by Adam.
>
> >"4-The performance of the proposed method is highly dependent on the polynomial interpolation degree K. I recommend an ablation study to investigate the effects of varying K to better understand its influence on model performance. In addition, comparisons with other node types beyond the LGR node leveraged in this study is also required."
>
> We discuss the impact of K in appendix I (G in the original submission). The higher the better typically in terms of numerical integration, but there are numerical challenges depending on data availability and noise in fitting high degree polynomials. On the choice of collocation method, a well known result from the literature is that there are 3 optimal choices in terms of collocation for ODEs as far as numerical error is concerned, Gauss Legendre, LGR and Legendre Gauss Lobatto and LGR is best suited for initial value problems.

---

> ### Comment · Reviewer_9Pjc · 2024-11-26
> **Thank you for your response**
>
> Thank you for your response.
>
> 1. The paragraph "Continuity constraints between subintervals" in Section 3.2 appears to be incomplete. It seems that this section needs to be rewritten to provide a complete explanation. While the ODE is Markovian, the discrete nature of the observations during learning can lead to discontinuities, and I believe a more detailed discussion is needed to clarify this point.
>
> 2. Additionally, upon reviewing the revised manuscript, I noticed that the changes were not clearly marked, and no indication was provided regarding where the revisions were made. This has made it difficult for me to locate the specific sections that have been updated.
>
> 3. I noted that there are still incorrect claims regarding the application of the Augmented Lagrangian method, which was not actually employed in the study. It seems there has been a misrepresentation of the methods used in the work. I believe it is crucial to correct this and accurately reflect the work conducted, preventing any potential misunderstandings.

---

> > ### Author Response · Authors · 2024-12-01
> >
> > We thank the reviewer for his time and attention.
> >
> > Point 2: We have uploaded a revision where colors show key modifications wrt the original submission: blue for edition and red for additions.
> >
> > Point 1: We have rewritten 3.2 on the continuity discussion and tied it in particular to the limitation section. The proposed method fits equations with interpolations. Even though data is shared across overlapping intervals, discontinuities of interpolations and intersections on overlaps are fatal for numerical reasons. As the method matches areas under the curve this error tends to be averaged in practice, but can be problematic if the integrated trajectory leaves the neighborhood of the data, hence the link to the limitation section.
> >
> > Point 3:
> > We have removed the reference to ADMM as it is not central to the paper and it was leading to confusion. The method is an alternate descent on different sets of variables and a separation of the objective function. Quadratic penalties as in the Augmented Lagrangian are used but multipliers are not updated.

---

### Official Review · Reviewer_n6sZ · 2024-11-03

**Soundness:** 3
**Presentation:** 3
**Contribution:** 3
**Rating:** 8
**Confidence:** 2

**Summary:**

The paper proposes a novel method to estimate parameters of (non-linear) ordinary differential equations. The methods is scalable and more efficient than existing approaches. The authors derive theoretical bounds on the difference between the optimal solution to the parameter estimation problem and the parameters found by optimizing the surrogate loss used by their method. The paper also evaluates the method on multiple benchmark problems and compares performances to several baselines and across different metrics.

**Strengths:**

- The paper is very clearly structured and well written which makes it easy to follow the ideas.
- The proposed idea is well motivated and positioned within the existing work
- The proposed methods seems to work well in practice on multiple problems and outperform existing approaches in terms of computational efficiency.
- I appreciate the detailed discussion of limitations.

**Weaknesses:**

- I appreciate the comparison with SINDy in the "Learning Polynomial dynamics" experiment, however, to my mind SINDy is not primarily a parameter estimation method but a way to learn the entire structure of the equation. For polynomials, estimating the coefficients of course implies learning the structure, nevertheless, I think it would be good to also include other parameter estimation methods in the comparisons in Table 2, e.g. BPTT, Adjoint methods and perhaps also other approaches such as simulation-based inference, as these (to my mind) would be the actual alternatives to the proposed approach.

- A minor general comment, I think the formatting is a bit off in some places so that lines of text are a bit to close to each other, e.g. between the caption of figure 2 and the main text or between table 3 and table 4.

**Questions:**

- In Figure 2, what do you mean by "observation error"?
- In Table 2, what is "IMATCH THRESH"?
- The true positive rate in figure 3 shows an initial drop as trajectories get longer before going up for all noise levels. Do you have any explanation for this (at least to me) surprising initial behavior?
- Not a question but rather a minor comment: In section 6, second paragraph, there seems to be something wrong with the sentence "If not, our method can be initialize parameters prior to adjoint methods". I didn't understand what this was supposed to mean.

---

> ### Author Response · Authors · 2024-11-21
>
> We thank the reviewer for his positive review and questions. We have uploaded a new version that improves the clarity of section 3.
>
> With regards to the Reviewer's questions:
>
> >"I appreciate the comparison with SINDy in the "Learning Polynomial dynamics" experiment, however, to my mind SINDy is not primarily a parameter estimation method but a way to learn the entire structure of the equation. For polynomials, estimating the coefficients of course implies learning the structure, nevertheless, I think it would be good to also include other parameter estimation methods in the comparisons in Table 2, e.g. BPTT, Adjoint methods and perhaps also other approaches such as simulation-based inference, as these (to my mind) would be the actual alternatives to the proposed approach."
>
> We will add those to future revisions.
>
> Questions:
> > In Figure 2, what do you mean by "observation error"?
>
> This is the signal to noise ratio. The figure illustrates that the non-uniform distribution of collocation points can create artifacts and amplify noise.
>
> In Table 2, what is "IMATCH THRESH"?
> > SINDy uses a sparse regression heuristic and the dynamic is sparse. IMATCH thresh is a variant that combines integral matching with sequential thresholding: upon convergence of integral matching, coefficients below a threshold in absolute value are forced to zero in all next iterations and the algorithm is rerun.
>
> > The true positive rate in figure 3 shows an initial drop as trajectories get longer before going up for all noise levels. Do you have any explanation for this (at least to me) surprising initial behavior?
>
> Features are highly correlated, this is especially complex on small trajectories. The RMSE on coefficients and on the derivative has a steady decline. We guess the initial drop can be an artifact due to sequential thresholding and correlations. In particular on Figure 3, there are 861 features that are highly correlated.
>
> >Not a question but rather a minor comment: In section 6, second paragraph, there seems to be something wrong with the sentence "If not, our method can be initialize parameters prior to adjoint methods". I didn't understand what this was supposed to mean.
>
> Thanks for catching this,  we have fixed it in the new version. The correct phrase was  "If not, our method can be used to initialize parameters prior to adjoint methods"

---

> > ### Comment · Reviewer_n6sZ · 2024-11-25
> > **Thank you for your answers**
> >
> > Thank you for the clarifications.
> >
> >
> > > This is the signal to noise ratio. The figure illustrates that the non-uniform distribution of collocation points can create artifacts and amplify noise.
> >
> > I find this highly confusing as the signal-to-noise ratio (SNR) is represented by the x-axis whereas the green line seems to represent the SNR by the y-axis, which actually represents the RMSE. I also do not quite understand why you would want to add a line for the SNR in the first place given that this is precisely what the x-axis represents.
> >
> > Moreover, higher values for the SNR usually correspond to a cleaner (=less noisy) signal but in case of Figure 2, the opposite seems to be the case, the RMSE is increasing with an increasing SNR.
> >
> > Lastly, I don't think representing the SNR in % is a good idea. What does it mean to have a SNR of (say) 10%? 10% of what? Similarly, I would not present the RMSE in %.
> >
> > All of these can probably relatively easily be fixed - but in its current version, I find figure 2 confusing / incorrect.

---

### Official Review · Reviewer_GuQS · 2024-11-03

**Soundness:** 3
**Presentation:** 2
**Contribution:** 3
**Rating:** 3
**Confidence:** 2

**Summary:**

This paper presents a novel method for estimating the parameters of an ordinary differential equation (ODE) by aligning a collocation-based estimate of the integral of the learned derivative with an interpolation of the trajectory. The problem addressed is significant, and the proposed approach appears to be innovative. However, certain sections of the paper lack clarity, as noted in the comments below.

**Strengths:**

The problem addressed is significant, and the proposed approach appears to be innovative.

**Weaknesses:**

However, certain sections of the paper lack clarity, as noted in the comments below

1. At the beginning of Section 3, the loss function 𝑙 is introduced, but this notation does not appear in Section 3.1. I assume it corresponds to the function in Equation (7)?

2. The matrices 𝐷 and 𝑉 in Section 3.1 are undefined. After Equation (4), it’s mentioned that the transformations are detailed in the appendix, yet the definition of \tilde{D} is still unclear.

3. The start of Section 3.2 states that (7) is solved using Algorithm 1. However, Algorithm 1 is written ambiguously, making it difficult to understand how it applies to solving (7). A more detailed, step-by-step description would be very helpful. For instance:

a. How is Θ initialized?
b. What exactly does Step 2, “build denoised set,” entail?
c. What is X_s in Step 6?

3.1 Additionally, it is unclear how Algorithm 1 is inspired by the ADMM algorithm (as claimed in Section 3.2); there appear to be no apparent connections between them.

4. The theoretical bound in Section 3.3 is not presented as a theorem. Without a precise statement of assumptions and conclusions, the main result of this paper is difficult to interpret. For example, the assumption in Section 3.3.1, “Supposing the gradient descent converges to an optimum Θ∗ such that L_2(Θ∗)=0” is unclear and appears overly restrictive in requiring convergence.

**Questions:**

See comments above.

---

> ### Author Response · Authors · 2024-11-21
>
> We thank the reviewer for his careful review that helps enhance the clarity and quality of the manuscript. We have uploaded a revised manuscript where we have modified section 3 (Pages 4,5,6), to clarify notations, provide more explanations while linking steps in the pseudo-code with the formulations and discussions.
>
> Specifically to the Reviewer's questions:
>
> >"At the beginning of Section 3, the loss function $\ell$ is introduced, but this notation does not appear in Section 3.1. I assume it corresponds to the function in Equation (7)?"
>
> The loss $\ell$ corresponds to the "system inversion terms" of the equation (6), or equivalently, the equation (5). This is made clearer in the revised manuscript.
>
> >"The matrices D and V in Section 3.1 are undefined. After Equation (4), it’s mentioned that the transformations are detailed in the appendix, yet the definition of $\tilde{D}$ is still unclear."
>
> We have explicited the general terms of matrices D and V in the revised manuscript. $\tilde{D}$ is the reordered matrix of constraints as in appendix D l.870. We also explicit that these notations are relatively common in the collocation literature (see Hager).
>
> >"a. How is $\Theta$ initialized ? b. What exactly does Step 2, “build denoised set,” entail? c. What is $X_s$ in Step 6?"
>
> We have modified our presentation in section 3.2 to link textual explanations with lines of the pseudo-algorithm, and equations whenever relevant. We have not discussed the initialization in section 3 as, the problem we obtain is in general a nonlinear nonconvex optimization problem that is solved by gradient descent. Initialization matters in general for convergence and for numerical stability for. ODE Solvers, just for convergence in our case. For the experimental section, we used the same initialization for the Neural ODEs experiments, and used $0$ for the other experiments, we have clarified this in the experimental section as well. The importance of the denoising step is described in section 3.2 and appendix A presents one strategy. We left it out of the main paper as it is not a contribution of our paper.
>
> >"3.1 Additionally, it is unclear how Algorithm 1 is inspired by the ADMM algorithm (as claimed in Section 3.2); there appears to be no apparent connections between them."
>
> We have added a section in the appendix to detail the connection. The motivation was to link the use of interpolation with alternate descent. We introduce a fictitious variable $Y$ with the equality constraint $X=Y$ and separate the two terms of the equation 7, (6 in the revised paper), one with variable $X$, the other with $Y$. Alternate descent or ADMM then arise from the relaxation of the equality constraint.
>
> >"The theoretical bound in Section 3.3 is not presented as a theorem. Without a precise statement of assumptions and conclusions, the main result of this paper is difficult to interpret. For example, the assumption in Section 3.3.1, “Supposing the gradient descent converges to an optimum $\Theta^*$ such that $L_2(\Theta^*)=0$” is unclear and appears overly restrictive in requiring convergence."
>
> Without additional hypothesis on $f$, the problem is nonconvex and there is no convergence guarantee. We thus propose two bounds, one asymptotic if $L_2(\Theta^*)=0$” and a second bound that depends on the value at any iteration. We have made this clearer in the revised manuscript. Experimental results show convergence, but should the method face limitations described in the limitations section such as numerical divergence, as the collocation method has its own error, the method is so fast compared to direct methods that it can still be valuable as the initialization phase of an ODE Solver method, to speed up convergence and avoid the initial plateau observed empirically as in Figure 1.

---

> > ### Comment · Reviewer_GuQS · 2024-11-27
> > **Thank you for your response**
> >
> > Thank you for addressing my questions and for the effort you put into clarifying the notations and improving the explanations.
> >
> > However, I noticed that the updated parts were not clearly marked in the revised version, making it challenging to identify the changes. Additionally, my main concern regarding the lack of a formal theoretical statement remains unaddressed.
> >
> > Lastly, I am still not convinced that the approach is closely related to ADMM. In my view, alternating descent differs significantly from ADMM, especially since the key step in ADMM—the multiplier update—is not used here.

---

> > > ### Author Response · Authors · 2024-12-01
> > > **New version**
> > >
> > > We thank the reviewer for his response. We have uploaded a new version where modifications compared to the original submission are highlighted in blue and additions in red so that the updated manuscript abides by the page limit.
> > >
> > > We have enhanced the formalization of the theoretical part.
> > >
> > > We have removed the references to ADMM as it is not central to the paper (it was mentioned as motivation in the original submission) and it was an apparent source of confusion. We never intended to claim properties of ADMM in particular. Without this mention, the method is presented as an alternate descent on two terms of the objective function.

---

### Meta-Review · Area_Chair_TnTZ · 2024-12-15

**Metareview:**

The paper introduces a new method for parameter identification in nonlinear ODEs using integral matching. This approach avoids the computational cost of adjoint methods and the noise sensitivity of gradient matching. Experiments on canonical systems and the Lorenz63 attractor show the method achieves significant speedups and better robustness to noise compared to existing techniques.

strengths: computational efficiency, robustness to observational noise, and scalability with system size. The experiments demonstrate speedups of up to three orders of magnitude over adjoint methods. The method is well-motivated and highlights limitations.

weaknesses: explanations in the theoretical section are said to be unclear and inconsistent. Experiments addressing continuity between subintervals are missing, and the method is not compared with all relevant alternatives, like BPTT or simulation-based inference.

While the idea is promising, the lack of clarity and incomplete validation make it unsuitable for acceptance at this stage.

**Additional Comments On Reviewer Discussion:**

Authors improved notations following reubtall and added explanations but left some inconsistencies. They revised the discussion on continuity, admitting it as a limitation but did not provide new experiments. The mention of ADMM was removed, addressing misrepresentation concerns. They promised additional metrics like RMSE on state variables.

---

### Decision · Program_Chairs · 2025-01-22

Reject